# Professional decision making with digitalisation of patient contacts in a medical advice setting: a qualitative study of a pilot project with a chat programme in Sweden

Åsa Cajander ,[1] Gustaf Hedström,[2] Sofia Leijon,[3] Marta Larusdottir[4]

¹Department of Information Technology, Uppsala Universitet Teknisk-naturvetenskapliga fakulteten, Uppsala, Sweden
²Department of Public Health and Caring Sciences, Uppsala Universitet, Uppsala, Sweden
³Department of Information Technology, Uppsala Universitet, Uppsala, Sweden
⁴School of Computer Science, Reykjavik University School of Computer Science, Reykjavik, Iceland

**Correspondence to**
Dr Åsa Cajander;
asa.cajander@it.uu.se

## ABSTRACT

**Objectives** Patient e-services are increasingly launched globally to make healthcare more efficient and digitalised. One area that is digitalised is medical advice, where patients asynchronously chat with nurses and physicians, with patients having filled in a form with predefined questions before the chat. This study aimed to explore how occupational professionalism and the possibility of professional judgement are affected when clinical patient contact is digitalised. The study's overall question concerns whether and how the scope of the healthcare staff's professional judgement and occupational professionalism are affected by digitalisation.

**Design and setting** A qualitative study of healthcare professionals working in a pilot project with a chat programme for patients in a medical advice setting in Sweden.

**Participants and analysis** Contextual inquiries and 17 interviews with nurses (n=9) and physicians (n=8). The interviews were thematically analysed. The analysis was inductive and based on theories of decision making.

**Results** Three themes emerged: (1) Predefined questions to patients not tailored for healthcare professionals' work, (2) reduced trust in written communication and (3) reduced opportunity to obtain information through chat communication.

**Conclusions** The results indicate that asynchronous chat with patients might reduce the opportunity for nurses and physicians to obtain and use professional knowledge and discretionary decision making. Furthermore, the system's design increases uncertainty in assessments and decision making, which reduces the range of occupational professionalism.

## INTRODUCTION

The idea that new technology will solve current or future difficulties at work is not new. The digitalisation of healthcare is no exception when new technologies such as Artificial Intelligence have been highlighted as solutions to existing problems.[1 2] However, follow-ups have shown that the systems rarely deliver the expected benefits regarding,

> **Strengths and limitations of this study**
>
> ► A methodological strength of the study is the combination of field studies with an interview study.
> ► Another methodological strength is that this case study is based on an implementation project and not a research intervention.
> ► A further strength of the study is the unique and highly relevant research question that has not been explored before.
> ► A limitation of the study is the limited numbers of nurses and physicians in the interview study.

among other things, increased efficiency and safety.[3–6] Moreover, the results from eHealth implementations vary depending on the solution and organisational context.[7]

One recent area with varying implementation results is asynchronous communication platforms where healthcare professionals and patients interact.[8–12] For example, Jhala and Menon[8] described implementing an asynchronous communication platform for healthcare professionals to save time compared with synchronous communication. Moreover, Tran *et al* explained that primary care providers requested more than a generic approach to a common clinical scenario when they contacted a specialist via an e-consultant application designed for interaction between healthcare professionals.[10] Written asynchronous communication between healthcare professionals and patients during treatment was experienced in favourable terms by the patients regarding freedom, security and certainty. This was closely linked to patients stressing the advantages that the healthcare professionals know who they are. The patients also point to the importance of being familiar with the healthcare professionals.[9] Entezarjou *et al*[13] explored the

BMJ

staff's experience of an asynchronous chat for healthcare professionals and patients. Several benefits of the system were described, such as an increased opportunity for the patients to express themself as well as aid in triage by giving an overview of incoming presenting symptoms. For example, concerns about the system included the staff feeling involuntarily responsible for irrelevant symptoms reported by the platform and being overwhelmed with information for clinical decision making.[13] In addition, the implementation affected practice organisation and working environment. It can become problematic when information technology (IT) systems are developed without considering the skills used by those involved in patient care.[11] Other studies have also focused on the work engagement aspects of asynchronous chat systems. In one study, nurses experienced less time- and emotional pressure and a decline in job control and feedback from coworkers working from home.[12] when implementing such a system.

How professions should be differentiated from other occupations has long been debated.[14 15] Burrage *et al* presented a definition of professionalism in the 1990s,[16] which was further developed in the early 2000s[17]: There is often a high degree of science in knowledge development for professionals, but professional practice can also be based on other knowledge, called proven experience. Moreover, professional practice is often characterised by independence; it is the professional who decides on adequate actions and has significant influence over his division of work. Traditionally, the professions have been challenging to govern, and the typical control has taken place by the members themselves collegially. The possibility of sanctions, for example, exclusions, exists as disciplinary measures.

Evetts distinguishes organisational professionalism and occupational professionalism.[18] Occupational professionalism represents the classic image of professional management, including collegial authority, self-government and collegial review.[18] However, occupational professionalism also includes space for manoeuvre for one's assessments and decision making, so-called discretionary decision making. Moreover, control takes place within one's professional group and occupational professionalism is based on collegial authority with professionals in control of the work. This is in contrast to the so-called organisational professionalism. Instead, this professionalism focuses on legal-rational decision making, where tasks tend to be standardised by regulatory systems and a low degree and opportunity for discretionary assessment/decision making. In the organisational professionalism paradigm, there are hierarchical structures regarding authority and a striving for managers' power in the organisation. In this way of organising work evaluations, goals and goals fulfilment are common. Evetts further points out that occupational professionalism and organisational professionalism are two ideal types and that the reality is not as pure as the models suggest. Evetts clarifies that organisational professionalism might threaten and undermine occupational professionalism.[18 19] There is a risk that professional values and norms are subject to market logic and commercial principles, resulting from increased bureaucratisation, reduced opportunities for professional ethics and decreasing opportunities for discretionary decision making. In Evett's descriptions, trust, discretionary decision making and competence are central to professional execution.[18–20] Other researchers agree that discretionary decision making is major to professionals in their performance of tasks.[21 22] Although a high degree of discretionary decision making can be burdensome for professionals, this type of decision making enables the exercise of professional judgement when organisations' rules are vague, contradictory or even harmful to clients/patients.[23]

Even if technical innovations and automation are introduced in healthcare, the core of the patient-centred work for the care staff remains, including tacit knowledge, professional reflective practice and professional autonomy.[24–26] However, little is known about how increased automation in patient contact affects the healthcare staff's professional autonomy and decision making.

Hence this study aims to understand how occupational professionalism and the possibility of professional judgement are affected when clinical patient contact is digitalised. A project where nurses and physicians used a chat programme for communication with patients is the basis, and we analyse professional judgement and professionalism with the use of professionalism theory. The study's overall question concerns whether and how the scope of the healthcare staff's professional judgement and occupational professionalism are affected by this digitalisation.

## METHODS
### Contextual setting
In 2003, the national service 1177 was established, initially providing online self-care information and healthcare advice via telephone from nurses 24 hours a day, all year round. Around 1200 nurses work full or part time, assisting the 7 million people who call each year (from the 10 million inhabitants in Sweden). Approximately 30%–35% of all calls are given self-care advice, whereas the rest are referred to a healthcare facility.

Recently, politicians in Uppsala decided to introduce a digital system to facilitate citizens' contact with healthcare providers. A proof-of-concept project was started in 2019, enabling asynchronous chat between citizens and nurses working at 1177 in Uppsala. The chat is operated by one to two nurses during the daytime and evenings. In addition, a physician is available during the last four staffed hours each day. The number of advice-seekers has increased during the project but not met the expected number.

The IT system in use obliges the patients to fill in a predefined questionnaire aimed at their specific chief complaint from a list of available alternatives. The answers

are presented to the nurses and physicians. When the patient has filled in the form, the healthcare personnel interact with the patient via the asynchronous written chat function.

The work of the nurses and physicians in the project differ in some aspects. The nurses are accustomed to giving medical advice in a human-to-human setting using a telephone. To aid in their work, they have an electronic decision support system with which they are familiar. Physicians' primary duty in the project is to provide medical advice, diagnose and treat the patients. If neither were deemed possible, the physicians would refer patients to a physical healthcare setting.

### Qualitative methods

In this qualitative study, contextual inquiries[27] were conducted with five informants during 1 day to understand their work better. The researcher asked questions on how the work was done and took notes. The contextual inquiries were used to understand the context of the work and as background information.

This was followed by 17 semistructured interviews (25–60 min) conducted with nurses (n=9) and physicians (n=8; see online supplemental file 1 for the interview template). All participants from the contextual inquiries and the interviews were asked to provide written informed consent.

A professional transcriber did transcriptions verbatim, and the authors analysed the transcribed interviews as described in Clark and Braun's thematic analysis method.[28] We used a theoretical framework of occupational professionalism as presented in the introduction for the interpretation of the results.

### Patient and public involvement

This study did not include any patient or public involvement.

### RESULTS

Nine interviews with nurses were conducted. The gender distribution was eight females and one male. The age range was 29–64 years, with an average age of 49 years. The average number of years as a professional nurse was 21.1 years (range 5–36 years). The number of years of working experience at 1177 was on average 3.4 years (range 1–11 years).

Moreover, four female physicians and four male physicians were interviewed in the age range of 26–49 years, with an average age of 34 years. Professional experience was two specialists and six residents in general practice. No one had previous experience of having worked in 1177.

### Predefined questions to patients not tailored for healthcare professionals' work

Both nurses and physicians express concerns about the predefined questionnaires for the patients to fill in before communicating with the healthcare personnel via the asynchronous chat. The interviewees said that these questions were not the same questions as in the elderly telephone system, but were provided by the IT company and adapted to the 1177 context in workshops and were updated during the pilot project. The system automates these questions. The healthcare personnel cannot see what questions the system asks the patients and how they are phrased. They experience that the predefined questions to patients is not tailored for their work as they can only see the answers from the patients. When the patients have started the asynchronous written chat, only later in the contact comes an opportunity to ask the questions they consider relevant to the health issues problems that the patients describe. However, several nurses experience the lack of knowledge about how questions are phrased created uncertainty about how they should interpret the answers and what further questions to ask.

> And then we can't see what the questions are in the questionnaire, so what exactly should I ask? What, what questions do I ask with this new questionnaire? (Nurse 2).

The nurses and physicians express that the questionnaires are not perceived as tailored and adapted to healthcare counselling activities or triaging or determining patients' treatments' priority but are generally directed at healthcare. Consequently, many of the questions are perceived as irrelevant.

> It is not developed and directed for our needs, but may generally work for healthcare. Eh… so it is for example that eh… many questionnaires that the patient can choose when they seek care are not relevant to us. (Nurse 2).

The physicians also express different views within the medical profession regarding what is considered possible in digital care. Sometimes, it seems as if others think they should take care of more than what is possible.

> …it is probably Sweden's municipality and county council that has come up with a list of what you can handle via video and chat. (Physician 4).

> …the medical profession is quite divided about what can be diagnosed, such as a bacterial sore throat. Eh, I think it would be tough to judge based on a written chat. I would say almost impossible. (Physician 4).

### Reduced trust with written communication

The nurses express difficulties in trusting what patients state in text and checkboxes. They relate this to a more complex evaluation and understanding of the information compared with telephone contact. Also, some nurses claim that the instruction is not to ask follow-up questions since this is time-consuming and should base their judgement on the information provided in the written chat. However, they feel that even though they need to ask more follow-up questions, uncertainty can remain.

But I think it's because we're so insecure. We want more information before we make an assessment. We are not confident enough in the assessments we make. Because we want to ask more follow-up questions as we do on the phone. (Nurse 6).

One example highlighted by the nurses was that the forms provide information that they must consider, such as chest pain, even if that was not what the patient described as the cause of the contact. They describe uncertainty regarding this information since there is no information about when the symptoms occurred. It thus becomes difficult for nurses to assess the severity of the symptoms.

…which you have to ask about; 'is it true that you have palpitations and how long have you had it?' And then there may be something like 'Yes, I've had it for two years, it's nothing new', but I do not see it. And must take it very seriously. (Nurse 9).

There is an expressed desire among the nurses to influence which questions the patient receives in the questionnaires significantly. Therefore, the nurses should have an opportunity to choose for themselves how and which questions to ask.

In comparison to the nurses, several physicians express a positive perspective regarding work and nurses triaging patients. The physicians feel they receive much information before contacting the patient, which is an effective way to work. However, they also experience it negatively to have very little influence over the questionnaires when there are shortcomings.

It's also a thing like this when you have eh… the cause of cough for example. And nowhere is there the question of whether the patient is a smoker. (Physician 1).

### Reduced opportunity to obtain information through chat communication

Several nurses describe aspects that are addressed that are lost in a chat contact compared with the telephone: Examples are the ability to hear any effect on the patient's breathing. They describe information such as hearing if the patient's breathing is heavy, hissing, strained or mushy speech, which could guide the assessment of ailments for counselling. Aspects such as the occurrence and differentiation of dry cough or mucus cough are also present in this. For example, one nurse describes that she wants to hear a person's breathing to assess whether the situation is urgent. The nurses also discuss the patients' estimates of pain and their information to interpret these estimates. For example, patients might state a very high level of pain on the phone, but being unaffected in the voice may not have severe pain as reported by the patient and not urgent with emergency interventions.

You sound very unaffected in the voice and things like this and so on and then it becomes like this, then

I can still hear 'no but it's probably not as serious as the patient states. (Nurse 1).

This type of information is described as disappearing when working with the chat in comparison with a telephone. As a result, there is only the patient's graded pain in the written conversation, and it becomes more difficult for the nurses to make emergency assessments.

Both nurses and physicians also compare the possibility of making an assessment based on information from the chat and information available at a physical patient visit. For example, nurses describe that a physical patient visit allows controlling vital parameters such as blood pressure, heart rate, respiration. However, this is not possible when they chat with the patients. Therefore, several nurses describe the physical visits, meet and see the patient, and do the previously mentioned examinations and checks to be the best alternative for assessing healthcare.

The physicians' stories describe limitations and difficulties in their assessments based on a chat contact. Physicians perceive it difficult when they do not receive answers to their questions. If they receive responses, they become uncertain what to do with answers that are not relevant based on the patient's search cause but potentially may be medically appropriate. In the chat, the assessment basis becomes what the patient chooses to tell. Physicians express that it is more difficult to determine whether all relevant information is provided or not in the chat. They compare these situations to physical visits where they can depend on both the conversation and the physical examination.

So, it is also the patient's responsibility to a certain extent. I mean, I can only judge from what you say to me over the chat and I can only judge from the outside as physical and what you get in the conversation in the examination room. What happens in 24 hours, I do not know? (Physician 7).

It appears that physicians see the chat as a compliment in many ways but not able to replace physical visits fully. Physicians express that it is only a small number of conditions that they can assess with certainty. Compared with physical visits, much of the information they need in their assessments and decision choices disappear in the chat.

My opinion is that video and chat can't replace a physical visit with the same high quality. There are so many parameters that can make the quality and assessment better with a physical visit. (Physician 4).

Like the nurses, physicians describe aspects such as assessing how a person speaks and moves. Some parameters can be small but crucial to warn that something is not right and not perceived in a chat conversation. Also, the information in the text becomes more difficult to interpret when it comes to specific symptoms.

But I think that like it is much harder to understand a stomach with just words like this 'stomach ache' than if they were sitting in front of you and pointing or like

more explained because it should be written a lot if you are to understand how the stomach ache because it can be so incredibly different while yes but fever, fever is a fever after all. (physician 6).

## DISCUSSION

Even though both nurses and physicians think the predefined forms provide much information before meetings, there are different opinions if they are time-saving or not. These different opinions could be due to different approaches in dealing with the current situation. Both nurses and physicians describe that they have little insight into how and which questions are asked to patients via the predefined forms. Hence, they have two alternatives; rely on the system, although they express that the information given is not of the same quality as the information provided in a conversation, or repeat the majority of questions since an interpretation of an answer without knowing the question is difficult, which is the case with the predefined forms. They also said that the forms are not aimed at their activities, and they felt that they had minimal opportunity to influence the content of the forms. The problem with little contact between the IT department and the healthcare professionals is well-known[29 30] and is a root cause of this problem.

An IT system forcing healthcare staff to perform the work within a more limited framework is an example of when occupational professionalism turns into organisational professionalism, as described in research [19 20 31]. They express increased organisational professionalism as healthcare professionals lose control and influence over essential elements of their professional practice and are expected to follow a standardised structure in which they have little insight.

The nurses have difficulties trusting what patients state in text and checkboxes. They experience they can make a more holistic evaluation and understanding of the information in a telephone conversation. Also, some nurses claim that the instruction is not to ask follow-up questions since this is time-consuming, and they should base their judgement on the information provided in the written chat. However, they feel that even though they need to ask more follow-up questions, uncertainty can remain. The issue of trust and automated systems is well-known in other areas. Research has shown that the professionals' trust increases if more information is provided by the system.[32 33]

One example highlighted by the nurses was that the forms provide information that they must consider, such as chest pain, even if that was not what the patient described as the cause of the contact. They describe uncertainty regarding this information since there is no information about when the symptoms occurred and their severity. It, thus, becomes difficult for nurses to assess the severity of the symptoms. This is also an example of these standardised forms and questions resulting in too much

information. This is consistent with previous research[34] that too much information overturns rather than helps in assessments and decisions. More data reveals a higher degree of complexity that the professional also needs to take into account. Klein[34] states that it is not always more or better information required to make professional decisions and reduce uncertainty. On the contrary, more intake can increase complexity and thus lead to more significant uncertainty.

The nurses described difficulties in assessments of potentially urgent cases. It is difficult to interpret and understand, for example, patients' estimates of pain or the occurrence of breathing difficulties. The physicians report problems in evaluating symptoms and limited opportunities to make reliable decisions. Some information they would need for this is not available more than in some specific and straightforward cases. Both nurses and physicians express that chat contact cannot correspond to a physical visit. Physicians describe perceived expectations from others that chat contact should be compared with a visit to a health centre, whereas nurses mainly relate to differences from telephone counselling.

At the same time, as the possibility for nurses and physicians' discretionary decision making decreases, they remain legally responsible for their choices. Who is accountable for the risks that result from the loss of information, and how much should healthcare professionals endure the uncertainty that a digital patient contact generates.

The scope for professional autonomy and discretionary decision making is reduced through the introduction of the chat system. Still, the advantage is that by adhering to predetermined procedures, professionals can protect themselves against possible criticism, and in this experience, feel more confident in their decisions. Standardisations can be seen to stand for a professional, scientific execution, free from morality and subjective interpretations, and in this, be more clearly customer or user oriented.[35] A disadvantage of this, in turn, is that professionals may be less likely to take risks in their professional practice: that it becomes more important to follow the standardisations that exist, with the consequence that professional values and professional independence are left behind.

**Twitter** Åsa Cajander @AsaC

**Contributors** ÅC and ML did the conception and design of the study. ÅC and ML did the data collection. GH and SL did the analysis and interpretation of the data. SL drafted the first version of the manuscript. ÅC, ML, GH and SL contributed to the interpretation, work with the finalisation of the manuscript and approved the final manuscript and agreed to be accountable for all aspects of the work in ensuring integrity and accuracy. The guarantor is ÅC.

**Funding** This work was supported by AFA grant number 180 250.

**Competing interests** None declared.

**Patient consent for publication** Not applicable.

**Ethics approval** The Swedish Ethical Review Authority ethically approved the study with the number 2019-04991.

**Provenance and peer review** Not commissioned; externally peer reviewed.

**Data availability statement** No data are available.

**ORCID iD**
Åsa Cajander http://orcid.org/0000-0001-7472-2215

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
