## [Reviewer comments · BMJ Open]

ARTICLE DETAILS

TITLE (PROVISIONAL)	Professional Decision Making with Digitalisation of Patient Contacts in a Medical Advice Setting: A Qualitative Study of a Pilot Project with a Chat Program in Sweden
AUTHORS	Cajander, Åsa; Hedström, Gustaf; Leijon, Sofia; Larusdottir, Marta

VERSION 1 – REVIEW

REVIEWER	Gavrilov, Goce University American College Skopje
REVIEW RETURNED	13-Jul-2021

GENERAL COMMENTS	no comments
-------------

REVIEWER	Koch, Sabine Karolinska Institutet
REVIEW RETURNED	30-Sep-2021

GENERAL COMMENTS	I accepted to review the paper as it caught my interest. After reading the abstract, I assumed the objective of the paper would be to assess the effects of asynchronous chat on professional decision making. However, after reading the paper, I have a number of questions. My major criticism is related to the lack of contextual description and the methods which are not detailed enough. I elaborate on my comments below according to the different sections of the paper. Abstract: The text under objectives is rather a background description than an objective. The last sentence " However, little is known about the effects on professional decision making and assessment, which are parts of professionalism theories." suggests, however, that assessing the effect of asynchronous chat on professional decision making might be the objective. In the introduction the authors state another objective though i.e. to answer the overall question whether and how the scope of the healthcare staff's professional judgment and occupational professionalism are affected by digitalisation". This objective is quite broad and should in my mind be more targeted towards the specific study. In any case the objective in abstract and introduction should match.
---

	Participants, Setting, and Primary and secondary outcome: I am sure the authors had to follow a template by the journal here, but asking for primary and secondary outcome is more suitable for a clinical study. Having a simple "Methods" section instead, where the authors can detail their study setting including participants and the methods used, including methods for data analysis would in my opinion be more suitable for this type of paper. Conclusions: I think the conclusion that "asynchronous chat with patients reduces the opportunity for nurses and physicians to obtain and use professional knowledge and discretionary decision-making" cannot be drawn in general from the limited data. This depends on the respective context in which asynchronous chat is used, as well as the design of the system. Minor: p.4 line 8: "asynchronous" should be an adverb Introduction: -Some minor spelling mistakes (e.g. line 39 – "this study" should be "one study" as it refers to [12] and not the current manuscript) - broad objective (see comment in abstract) Section 2 about professionalism gives a good overview about the theoretical framework that the authors aim to use for the analysis of the results but it could be shortened in favor of a more detailed methods and context description that would enhance interpretability of the results for the reader. Case study - This section is too short. I think the issue I have with interpreting your results is very much related to an insufficient description of the context. You are not looking at the changes between a telephone work process in 1177 compared to an asynchronous chat, but you base it on an asynchronous chat with preceding questionnaire that is known to the patients but not to the staff. Different questions are issued based on the chief complaint the patient presents with but do these questions resemble e.g. the decision-support system (SRÅ) that 1177 staff also used conventionally or is this something new? Who developed the questions? How was the process for that? E.g. in the conclusions you write that insufficient collaboration between the IT department and clinicians and clinicians' insufficient influence is a problem. This is true in general, but was it so in this specific case study? I doubt that the IT department had any influence on the provision of the questions. It is a bit difficult to see how you draw conclusions based on your results. Methods Please describe in more detail how the contextual inquiries were done. Who? How many? Time duration? How were the participants (also for the interviews) recruited? It seems e.g. that the doctors that were interviewed had never worked in this role. Also add that you used the theoretical framework for the interpretation of your results. This is not obvious from your methods description. I also think the interview guide should be added. The authors state that they have ethical approval, but do not write whether informed consent was taken from the participants. Results
--	---

	The results as such are relatively clear but having the interview guide would help, and also some results from the contextual inquiry. It seems that all results come from the interviews? Discussion: As the objective is quite broad, and the context unclear, I would not say that the discussion and conclusion can be justified by the results. The major discussion points relate to the fact that questions (unknown to the physician) are issued, which is not really related to an asynchronous chat but to the context in which it is used. Language: The language is ok in general but the authors should read through the text and correct some mistakes, e.g. mixing up singular/plural and the use of some incorrect translations from Swedish. SKL = SKR should be SALAR & the official English translation. Even if participants used Swedish abbreviations, authors should use the correct English ones, and add the respective explanation in square brackets; "contact cause" or "search cause", which the authors also used, is not the correct concept in English – I assume the authors mean "chief complaint"
--	---

VERSION 1 – AUTHOR RESPONSE

Reviewer 2 Comments to the Author:

11. The text under objectives is rather a background description than an objective. The last sentence " However, little is known about the effects on professional decision making and assessment, which are parts of professionalism theories." suggests, however, that assessing the effect of asynchronous chat on professional decision making might be the objective. In the introduction the authors state another objective though i.e. to answer the overall question whether and how the scope of the healthcare staff's professional judgment and occupational professionalism are affected by digitalisation".

This objective is quite broad and should in my mind be more targeted towards the specific study. In any case the objective in abstract and introduction should match.

Response: We have improved the text about the objective of the study and also seen to it that the text in the abstract and the introduction is the same.

12. Participants, Setting, and Primary and secondary outcome: I am sure the authors had to follow a template by the journal here, but asking for primary and secondary outcome is more suitable for a clinical study. Having a simple "Methods" section instead, where the authors can detail their study setting including participants and the methods used, including methods for data analysis would in my opinion be more suitable for this type of paper.

Response: We have changed the abstract and taken out primary and secondary outcomes, and added a methods section.

13. Conclusions: I think the conclusion that "asynchronous chat with patients reduces the opportunity for nurses and physicians to obtain and use professional knowledge and

discretionary decision-making” cannot be drawn in general from the limited data. This depends on the respective context in which asynchronous chat is used, as well as the design of the system.

Response: We have added “might” to the conclusion to indicate that the result is tentative.
14. p.4 line 8: “asynchronous” should be an adverb.

Response: We have changed it to asynchronously.

15. Introduction: Some minor spelling mistakes (e.g. line 39 – “this study” should be “one study” as it refers to [12] and not the current manuscript)

Response: We have changed the wording.

16. broad objective (see comment in abstract)

Response: We have improved the description of the objective and clarified it a bit.

17. Section 2 about professionalism gives a good overview about the theoretical framework that the authors aim to use for the analysis of the results but it could be shortened in favor of a more detailed methods and context description that would enhance interpretability of the results for the reader.

Response: We have shortened the section and taken out the table, we have also added some sentences to the method to clarify.

18. Case study: This section is too short. I think the issue I have with interpreting your results is very much related to an insufficient description of the context. You are not looking at the changes between a telephone work process in 1177 compared to an asynchronous chat, but you base it on an asynchronous chat with preceding questionnaire that is known to the patients but not to the staff. Different questions are issued based on the chief complaint the patient presents with but do these questions resemble e.g. the decision-support system (SRÅ) that 1177 staff also used conventionally or is this something new? Who developed the questions? How was the process for that? E.g. in the conclusions you write that insufficient collaboration between the IT department and clinicians and clinicians’ insufficient influence is a problem. This is true in general, but was it so in this specific case study? I doubt that the IT department had any influence on the provision of the questions. It is a bit difficult to see how you draw conclusions based on your results.

Response: We added some text about the questions asked to the patient and the process when they were developed, but we put this text under results as the description merely is based on what the nurses and physicians mentioned in the contextual interviews and the interviews.

19. Methods: Please describe in more detail how the contextual inquiries were done. Who? How many? Time duration? How were the participants (also for the interviews) recruited? It seems e.g. that the doctors that were interviewed had never worked in this role.

Response: We added some clarification in the methods section, and also added that the contextual interviews were used to get an understanding of the context.

20. Also add that you used the theoretical framework for the interpretation of your results. This is not obvious from your methods description.

Response: We have added a sentence on that in the Method section.

21. I also think the interview guide should be added.

Response: We have complemented the submission with the interview guide.

22. The authors state that they have ethical approval, but do not write whether informed consent was taken from the participants.

Response: We have clarified this in the methods section.

23. Results: The results as such are relatively clear but having the interview guide would help, and also some results from the contextual inquiry. It seems that all results come from the interviews?

Response: We added text about this in the methods section and clarified that the thematic analysis was done on the interview data only. The contextual inquiries were only used to understand the context.

24. Discussion: As the objective is quite broad, and the context unclear, I would not say that the discussion and conclusion can be justified by the results. The major discussion points relate to the fact that questions (unknown to the physician) are issued, which is not really related to an asynchronous chat but to the context in which it is used.

Response: We have added some words to the description of the objective to make it clearer, and then the discussion is appropriate related to that objective.

25. Language: The language is ok in general but the authors should read through the text and correct some mistakes, e.g. mixing up singular/plural and the use of some incorrect translations from Swedish.

Response: We have sent the paper to a professional copy-editing service and improved the language throughout.

26. SKL = SKR should be SALAR & the official English translation. Even if participants used Swedish abbreviations, authors should use the correct English ones, and add the respective explanation in square brackets; "contact cause" or "search cause", which the authors also used, is not the correct concept in English – I assume the authors mean "chief complaint".

Response: We took the abbreviation out as it only confuses the reader and instead used the description of what SKR is.

VERSION 2 – REVIEW

REVIEWER	Koch, Sabine Karolinska Institutet
REVIEW RETURNED	06-Nov-2021
GENERAL COMMENTS	The authors have addressed all my previous comments satisfactorily. I only changed minor typos in the attached file.